# Cork Porous Biocomposites with Polyurethane Matrix Modified with Polyol Based on Used Cooking Oil

**DOI:** 10.3390/ma16083032

**Published:** 2023-04-12

**Authors:** Maria Kurańska, Mariusz Ptak, Elżbieta Malewska, Aleksander Prociak, Mateusz Barczewski, Mateusz Dymek, Fábio A. O. Fernandes, Ricardo Alves de Sousa, Krzysztof Polaczek, Karolina Studniarz, Katarzyna Uram

**Affiliations:** 1Department of Chemistry and Technology of Polymers, Cracow University of Technology, Warszawska 24, 31-155 Cracow, Poland; maria.kuranska@pk.edu.pl (M.K.); aleksander.prociak@pk.edu.pl (A.P.); krzysztof.polaczek@doktorant.pk.edu.pl (K.P.); karolina.studniarz@student.pk.edu.pl (K.S.); katarzyna.uram@doktorant.pk.edu.pl (K.U.); 2Faculty of Mechanical Engineering, Wroclaw University of Science and Technology, Łukasiewicza 7/9, 50-371 Wrocław, Poland; mariusz.ptak@pwr.edu.pl (M.P.); mateusz.dymek@pwr.edu.pl (M.D.); 3Institute of Materials Technology, Poznan University of Technology, Piotrowo 3, 61-138 Poznan, Poland; mateusz.barczewski@put.poznan.pl; 4TEMA: Centre for Mechanical Technology and Automation, Department of Mechanical Engineering, University of Aveiro, 3810-193 Aveiro, Portugal; fabiofernandes@ua.pt (F.A.O.F.); rsousa@ua.pt (R.A.d.S.); 5LASI—Intelligent Systems Associate Laboratory, 4800-058 Guimaraes, Portugal

**Keywords:** biocomposites, polyurethane, cork, polyol

## Abstract

Renewable materials are materials that are replenished naturally and can be used again and again. These materials include things such as bamboo, cork, hemp, and recycled plastic. The use of renewable components helps to reduce the dependence on petrochemical resources and reduce waste. Adopting these materials in various industries such as construction, packaging, and textiles can lead to a more sustainable future and decrease the carbon footprint. The presented research describes new porous polyurethane biocomposites based on used cooking oil polyol (50 per hundred polyol—php) modified with cork (3, 6, 9, and 12 php). The research described here demonstrated that it is possible to replace some petrochemical raw materials with raw materials of renewable origin. This was achieved by replacing one of the petrochemical components used for the synthesis of the polyurethane matrix with a waste vegetable oil component. The modified foams were analyzed in terms of their apparent density, coefficient of thermal conductivity, compressive strength at 10% of deformation, brittleness, short-term water absorption, thermal stability, and water vapor permeability, while their morphology was examined using scanning electron microscopy and the content of closed cells. After the successful introduction of a bio-filler, it was found that the thermal insulation properties of the modified biomaterials were comparable to those of the reference material. It was concluded that it is possible to replace some petrochemical raw materials with raw materials of renewable origin.

## 1. Introduction

Currently, a large part of research is devoted to the search for functional materials from renewable resources that are biodegradable and completely safe for humans and the environment. The main reason for this trend is environmental protection. This is an essential aspect due to accelerating climate change. There are more and more legal regulations regarding the reduction in carbon dioxide emissions into the atmosphere and rational waste management. There is also a growing awareness of the consequences of production based on non-renewable resources, such as petrochemical raw materials, as well as the environmental pollution caused by poorly managed waste [1,2]. Household waste emits carbon dioxide and methane, contributing to climate change and polluting groundwater, soil, and air. One such waste example is used cooking oil [3].

Waste cooking oil (WCO) is defined as various types of vegetable oils recovered from food preparation, especially from the frying process. During frying, especially in large catering companies, the oil is heated several times. This process results in such oil having a high free fatty acid (FFA) content, making it unsuitable for consumption [4]. WCO causes many problems with their removal from the environment, as they contaminate water and soil. Instead of throwing away WCO and harming the environment, it can be used as a raw material for the production of biocomponents [5]. It is estimated that the price of WCO is 2–3 times cheaper than fresh vegetable oils [6]. The amount of waste cooking oil generated in individual countries varies and is difficult to estimate. In the European Union, approximately ~700,000–1,000,000 tons/year are recovered [7]. The successful conversion of the waste stream is the basis of a circular economy. Such an approach strikes a balance between industrial, economic, and environmental development [8].

Great attention is also paid to reducing energy consumption, particularly to preventing heat loss. This is achieved mainly by improving the thermal insulation of buildings, equipment, and pipelines in production halls. As a result of the increasingly stringent requirements for buildings in terms of energy efficiency, new materials with a low thermal conductivity coefficient are being sought. Such materials could be successfully used as the insulation of walls and roofs. Moreover, it is expected that the materials in question should be environmentally friendly and not contribute to the progress of climate change. For this reason, renewable raw insulation materials are becoming more and more popular [9]. In this context, vegetable oils, biomass plastic waste, and cork material are being intensively investigated [10,11,12,13].

Biofillers used in polymer composites also play an important role in environmental protection [14,15,16,17]. These materials of natural origin successfully replace conventional fillers, such as glass fibers, in many applications. Thanks to biofillers’ use in the production of polymer composites, their weight is reduced, their price is lower, and the consumption of petrochemical raw materials is reduced. Biofillers, mainly plant-based, improve many properties of polymeric materials, e.g., mechanical strength, at the level achieved with conventional inorganic and fossil fillers. The introduction of the most commonly used natural fillers in the form of natural fibers makes it possible to replace inorganic (glass and basalt) or synthetic fibers [18,19]. However, in many applications, including the production of foamed materials, the final composite structure’s effective impact cannot be achieved [20]. At the same time, the diversity of plant derivatives, resulting not only from the variable share of essential plant components such as lignin, cellulose, or hemicellulose but also other substances, including low-molecular-weight compounds, fats, or essential oils, makes the introduction of crushed plant parts such as particle-shaped bio-fillers a new prospect for shaping the properties of polymer composites [21,22].

Natural cork is a sustainable, renewable, and biodegradable material that has been used in engineering for centuries. Cork oak forests grow in the western Mediterranean and cover more than 2 million ha, especially in the southern Iberian Peninsula and northern Africa. It is estimated that in the 1990s, Portugal had about 713,000 ha, Spain had 475,000 ha, France had 68,000 ha, Italy had 65,000 ha, Morocco had 348,000, Tunisia had 90,000 ha, and Algeria had about 230,000 ha [23]. Portugal and Spain are the main producers of cork products. Annual cork production is around 374,000 tons. Portugal and Spain are responsible for 74% of the total production (51% and 23%, respectively) [24].

According to the Portuguese Cork Association, the cork industry generates over a billion euros in revenue annually and provides jobs for around 12,000 people. The cork industry supports a wide range of other businesses such as cork processors, designers, and manufacturers of cork products. Portugal has a 62.4% share of the world trade in cork, being the largest cork exporter in the world. The last Portuguese national forest inventory showed that the cork oak occupied approximately 23% of the forest area [25].

Cork is used for a variety of applications, ranging from heat insulation and soundproofing to decorative accents and even as a replacement material for traditional plastic and rubber components. Natural cork is composed of suberin, a waxy substance that makes it impermeable to moisture and air as well as resistant to mold, mildew, and bacteria. It is also lightweight, compressible, and resistant to fire and abrasion, making it an ideal material for engineering applications [26]. When used in engineering, natural cork has several advantages over traditional materials. It is environmentally friendly and renewable, making it an ideal choice for those looking to reduce their environmental footprint. Natural cork is also a sound and thermal insulator, making it a great choice for applications in which noise or temperature control is important [27]. Its natural properties make it an excellent choice for use in the construction, automotive, and electronics industries. Natural cork has also been used for various decorative applications such as wall coverings, door trims, flooring, and furniture. Its unique texture and color make it an ideal material for creating unique and interesting designs.

Cork is characterized by various intriguing qualities, including flexibility, great physical stability, compressibility, resistance to repeated long-term loads, and insulation from heat, electricity, and sound [23]. Natural cork’s low thermal conductivity coefficient, which ranges from 0.040 to 0.045 W/(m °C) [23], and its high acoustic insulation allow for a reduction in the sound intensity of 31 to 36 dB, depending on the material thickness [26]. Natural cork provides a barrier to liquids and gases, is chemically inert, and does not absorb odors when in contact with them [23,28,29].

The modification with cork of the polymer matrix can improve properties of the obtained biocomposites. Fernandes et al. prepared composites based on polypropylene (PP) or polyethylene (PE) and cork powder (50 wt.%). Cork was mixed with polymer by pultrusion and, next, samples were produced by compression molding. The density of the composites ranged from 1018 kg/m^3^ for the PP/Cork composites to 1061 kg/m^3^ for the PE/Cork. The flexural strength and strain were considerably reduced (samples with 50 wt.% of cork), and the modulus was improved in both cases. The modification of polyolefin by the addition of cork powder improved their energy absorption capacity [30]. Zur et al. prepared polyolefin foams characterized by increased tensile and compressive strength thanks to the modification with cork powder. The polymeric foam was modified with cork in an amount of 0.1 to 25 parts per hundred by weight of selected polyolefin. The foams described in the patent were characterized by a compressive stress of between 20 and 2000 kPa, a tensile strength of between 100 and 10,000 kPa, and an elongation at break of between 30 and 500%. The apparent density was in the range of 40–250 kg/m^3^ [31]. Oprea prepared polyurethane (PUR) resins modified with cork powder. Mechanical properties were examined in composites that had between 1 and 15% of cork and between 2 and 3% of carbon black. The type and amount of filler were found to significantly affect the mechanical properties of PUR materials. Typically, the modification of PUR with cork results in an increase in Young’s modulus and a decrease in elongation at break [32].

Research on the modification of rigid PUR foams with renewable raw materials is conducted by many scientists. However, the literature so far has not presented research on the modification of rigid PUR foams modified with both bio-polyol from used cooking oil and cork. This approach allows the increase in the total share of renewable raw materials in the PUR matrix. In this study, different contents of cork waste as well as bio-polyol from used cooking oil were used for modification of the polyurethane system. The main goal was to analyze the influence of cork bio-filler on the foaming process of the polyurethane system with and without the bio-polyol component, as well as on the cellular structure and selected properties of the obtained porous composites. In the reference polyurethane system, 50% of petrochemical polyol was replaced by bio-polyol synthesized from used cooking oil in order to increase the content of renewable raw material in final porous composites. The influence of bio-components on the usage of composite properties was discussed, taking into account changes in their cellular structure.

## 2. Experiment

### 2.1. Materials

Petrochemical polyol Rokopol RF-551 (hydroxyl value of 420 mgKOH/g and water content of 0.10 wt.%) was supplied by PCC Rokita S.A. (Brzeg Dolny, Poland). Bio-polyol (hydroxyl value of 370 mgKOH/g and water content of 0.05% mas) was obtained by a transesterification reaction of WCO with triethanolamine (Cracow University of Technology, Kraków, Poland). Polycat 9 produced by Evonik Industries AG (Essen, Germany) was used as a catalyst. Niax silicone L-6915 supplied by Momentive Performance Materials Inc. (Waterford, NY, USA) was used as a surfactant. TCPP from Lanxess AG (Cologne, Germany) was added to reduce PUR foams’ flammability. Ongronat 2100 was supplied by Borschodchem (Kazincbaricka, Hungary). The water was used as a chemical blowing agent. Cork was supplied by Amorim Cork Composites (Portugal). The grain size was approximately 1 mm and the density was equal to 200 kg/m^3^.

### 2.2. Preparation of PUR Foams and Their Composites with Cork

PUR foams were synthesized using a one-stage method from two-component systems, in which component A is the so-called polyol premix and component B is an isocyanate. The polyol premix consisted of a polyol or a mixture of a petrochemical polyol and a bio-polyol, a catalyst, a surfactant, a flame retardant, and water. The polyol premix was prepared in polypropylene containers with a capacity of 500 mL. Then, the appropriate amount of isocyanate was poured into the container with the premix and immediately stirred with a mechanical stirrer (RPM 1800) for 7 s. The composites were prepared with 3, 6, 9, and 12% of cork in relation to the polyol mass of the PU system. After this time, the composition was poured into previously prepared plastic molds, where the material grew and cross-linked. The materials were tested 24 h after the synthesis.

The reference materials were synthesized using two formulations: one based solely on petrochemical polyol (PU) and the other based on a mixture of petrochemical polyol and bio-polyol (BPU) in a 1:1 weight ratio. The isocyanate index of the synthesized materials was 1.1. Table 1 shows the quantitative compositions of both formulations per 100 g of the polyol components. Samples were labeled according to the system type (PU or BPU) and the cork content.

The quantities of ingredients in the formulations were chosen to keep the total amount of water in the system and the isocyanate index constant.

### 2.3. Characterization of Foaming Process and PUR Composites’ Properties

The influence of cork on the foaming process was analyzed using FOAMAT equipment. The device can record changes in the temperature, dielectric polarization, and pressure of reaction mixtures. The cellular structure was analyzed using a scanning electron microscope (SEM) TM3000 (Hitachi, Tokyo, Japan) and the software ImageJ (version 1.53f, U.S. National Institutes of Health, Bethesda, MD, USA) was used for SEM image analysis. The anisotropy index of foams was calculated as the major axis/minor axis ratio. The closed-cell content in the foams was measured according to the ISO4590:2016 standard. The thermal conductivity coefficient was measured at an average temperature of 10 °C with the use of a heat flow meter instrument Fox200 (TA Instruments, DE, USA) following the ISO 8301 standard. The foam samples had dimensions of 50 × 200 × 200 mm. The measurements were performed under the following conditions: hot plate temperature of 20 °C; cold plate temperature of 0 °C.

The apparent density and mechanical properties were measured according to the ISO 845:2006 and PN-EN 826:2013-07 standards, respectively.

The water vapor diffusion resistance factor (μ) of the PUR foams was determined according to the EN 12086:2013 standard. The test samples (cut out from the foam core) with a diameter of 94 mm and a thickness of 20 mm were sealed to the open side of a cylindrical test container with a desiccant—anhydrous calcium chloride. The air space between the desiccant and the test sample was approximately 20 mm. The test was conducted under a temperature of 23 ± 1 °C and relative humidity of 85 ± 3%. Six samples from each foam were tested. The weight of the test assembly was measured at 24 h intervals until the change in mass for each sample was within ±5% of the average sample value.

The brittleness test was carried out according to the ASTM C 421-08 standard and the result is presented as a percentage weight loss of the sample. Twelve cubic foam samples with 25 mm side lengths were placed in a standard oak box filled with 24 oak cubes having 20 mm side lengths. The box rotated around an axis at 60 rpm for 10 min. Brittleness was calculated according to Equation (1):(1)$$Brittleness=\left(\frac{m_{1}-m_{2}}{m_{1}}\right)\cdot 100\%$$
where *m*_1_ is the sample mass before the test in g; *m*_2_ is the sample mass after the test in g.

Dimensional stability (*DS*) was tested according to the ISO 2796-1986 standard. The specimens with dimensions of 100 × 100 × 25 mm were placed in a climate chamber at 70 °C and 90% relative humidity for 24 h. The DS was calculated using Equation (2):(2)$$DS=\left(\frac{l_{f}-l_{i}}{l_{i}}\right)\cdot 100\%$$
where *l_i_* is the initial length of the sample; *l_f_* is the sample length after the test.

Water absorption (*WA*) was measured according to the PN-93/C-89084 standard. The specimens with dimensions of 5 × 20 × 20 cm were immersed in distilled water for 24 h. Water absorption was calculated according to Equation (3):(3)$$WA=\left(\frac{m_{1}-m_{2}}{m_{2}}\right)\cdot 100\%$$
where *m*_1_ is the sample mass after absorption in g; m_2_ is the dry sample mass in g

The thermal properties of PUR were examined by thermogravimetric analysis (TGA) in the temperature range of 25 to 900 °C at a heating rate of 10 °C·min^−1^ under an inert nitrogen atmosphere using a TG 209 F1 Netzsch apparatus. The 10 mg ± 0.1 mg samples were placed in ceramic (Al_2_O_3_) pans.

The limiting oxygen index (LOI) test was performed based on the ISO 4589-2 standard. LOI indicates the lowest oxygen concentration (in volume %) in the mixture of oxygen and nitrogen necessary to sustain a stable combustion of a sample after ignition.

## 3. Results and Discussion

The synthesis of PUR foams is a highly exothermic process, and a large amount of heat is generated during the gelling and foaming reactions. The temperature is strongly correlated with the reactivity of a PUR system [33]. Using the FOAMAT device, the impact of the presence of bio-polyol and the cork content on the foaming process of PUR systems was examined. The influence of different contents of cork on the changes in dielectric polarization (a), temperature (b), and pressure (c) is shown in Figure 1.

Dielectric polarization is one of the parameters characterizing the reactivity of a system and it decreases as the reaction progresses. This is due to the gradual depletion of reactive free hydroxyl and isocyanate groups that form the urethane bonds. As shown in Figure 1a, the systems partially based on the bio-polyol were characterized by a faster decrease in dielectric polarization than the systems based solely on petrochemical polyol. A similar effect was observed in the work of Kurańska et al., where the influence of the bio-polyol content on the reactivity of PUR systems was also studied [3]. Most likely, this effect is caused by the presence of groups derived from a tertiary amine in the bio-polyol molecule, which has a catalytic effect. In addition, Figure 1a shows that the addition of a cork filler caused a slight decrease in the reactivity of the systems (in the case of the PUR systems not modified with bio-polyol), which may be due to the dilution of the reaction system. Cork contains hydroxyl groups in its structure, so there is a possibility of a reaction between them and isocyanate.

The foam core temperature also confirmed the systems’ higher reactivity based on the bio-polyol. In the case of the BPU systems, the temperature rise in the core of the foam was faster than in the systems based on petrochemical polyol (Figure 1b). The highest maximum temperature of the foam core was seen in the BPU/0 system. However, in the case of these filler-containing systems, the maximum temperatures were slightly lower, which also confirms the effect of cork on reducing the composition reactivity. No significant differences in maximum temperatures were observed for the systems based only on petrochemical polyol.

The pressure was determined as the pressure force of the reacting material on the surface of the measuring table. This measurement very well reflects the impact of the material on the mold walls during the synthesis of PUR foams. The short gelation time and the higher rate of foam growth favor lower pressures. Such a relationship can be observed for the BPU/0 foam, which is partly based on the bio-polyol and does not contain a filler [34]. On the other hand, in the case of filled foams characterized by lower reactivity, the pressure maxima are higher.

The cellular structure of PUR foams has a significant impact on their mechanical and thermal insulation properties. In order to determine the effect of the bio-filler content on the foam’s cellular structure, SEM micrographs of the samples were taken (Figure 2), and then the average cell surface area and anisotropy were analyzed (Table 2). The SEM micrographs presented here and their characteristics concern cells in a cross-section parallel to the direction of the material growth.

The introduction of cork into the PUR matrix resulted in an increase in cell dimensions, correlated with a decrease in the apparent density of the foamed PUR composites both with and without the bio-polyol. The effect of the bio-polyol on reducing cell dimensions was also visible compared to the materials based only on the petrochemical polyol. This is due to the presence of triglycerides of fatty acids, which have an effect typical of surfactants [35]. However, in the case of the BPU foams, the addition of cork caused an increase in cell dimensions, similarly to the PU foams. An exception in this trend among the PU foams was the PU/CM12 sample. This effect may be related to the increase in the apparent density of this material as compared to the foams modified with the bio-filler in the amount of 3, 6, and 9% by mass (relative to polyol premix). A partial replacement of the petrochemical polyol with the bio-polyol also increased the cell anisotropy coefficient, which is related to the higher reactivity of this system.

The percentage of closed cells had a significant impact on other foam properties, in particular, on thermal insulation properties. All the foams studied in our work had a closed-cell content of about 90% or more. A slight effect of the bio-polyol and modified cork on reducing the content of closed cells was observed.

The apparent density of rigid PUR foams significantly impacts their physical and mechanical properties. One of the objectives of this research was to obtain materials with an apparent density of about 45 kg/m^3^ (Table 2). Such a value of apparent density ensures appropriate physical and mechanical properties of foams. The foams containing modified cork had a lower apparent density compared to the PU/0 and BPU/0 reference materials. This effect is expected and beneficial from an economic point of view. Cork has a porous structure and is characterized by a low bulk density of 199.1 kg/m^3^ for modified cork [32]. Therefore, introducing it into a PUR composition reduces the apparent density of foams. However, no effect of the bio-polyol on the apparent density of the foams was noted in our work.

Conducting measurements of the thermal conductivity coefficient allowed the determination of the impact of the presence of the bio-polyol and cork content on the thermal insulation properties of the foams.

The thermal conductivity coefficient of the modified foams was comparable to that of the cork-free foam. The studies described in the literature show that the introduction of a filler increases the thermal conductivity coefficient. In this case, the lack of deterioration of the thermal insulation properties may be due to the good insulation properties of the cork. Low thermal conductivity is correlated with a high content of closed cells. The high content of closed cells means that the foaming gas, characterized by a low thermal conductivity coefficient, is trapped inside the material. This gas’s share of thermal conductivity in the total value of the λ coefficient is about 60–80% [3].

Table 3 presents the results of the water absorption test of the foams. The water absorption of rigid PUR foams is an important property when the foams are used as thermal insulation materials because the presence of water in such materials increases thermal conductivity, which is an unfavorable effect. Therefore, it is important that foams have low water absorption, below 3%.

Generally, all the foams tested in our experiments were characterized by a low water absorption. There was a slight increase in water uptake for the foams containing cork compared to the reference foams. This is most likely due to the interactions between the bonds in the cork structure and the water. The bio-polyol-based foams filled with cork partially exhibited a slightly lower water absorption than the petrochemical polyol-based foams. The water absorption of the materials was lower than 1.5%, which is considered a favorable value.

The water vapor diffusion resistance coefficient determines how high the resistance to water vapor penetration through a material is. The higher the value of this coefficient, the less permeable to water vapor the material is. Low water vapor permeability in the case of materials used as a thermal insulation layer can cause a disadvantageous moisture retention effect in the building. Commercially available closed-cell PUR spray foams are characterized by the coefficient µ in the range of 50–150. This means that these materials are characterized by medium vapor permeability. On the other hand, commercial open-cell foams are highly vapor-permeable and characterized by a value of the µ coefficient below 15. The foams obtained in our work were characterized by the water vapor diffusion resistance coefficient in the range of 13–21, making them well permeable to water vapor. The introduction of the bio-polyol to the PUR matrix resulted in a slight increase in the µ coefficient. The cork introduced into the compositions increased vapor permeability.

Brittleness is another important parameter characterizing the properties of PUR foams. This property is a measure of the tendency of a given material to lose weight due to abrasion under the influence of mechanical factors. The brittleness of the foams produced in our experiments was very low, and the mass loss of the materials did not exceed 3%. The introduction of a filler in the form of cork was beneficial as it reduced the foams’ brittleness. There was no significant effect of the presence of the bio-polyol on the brittleness of the rigid PUR foams. The foams were characterized by slightly changing dimensions not exceeding 1.5%. No significant influence of the presence of the bio-polyol or cork content on dimensional stability was observed.

Compressive strength measurements were carried out both in a direction parallel and perpendicular to the direction of the foam’s growth (Figure 3).

The differences in the results of compressive strength measurements in the parallel and perpendicular directions reflect the anisotropic nature of the cellular foam structure. The compressive strength in the parallel direction was higher compared to the perpendicular direction, which is related to the shape of the cells. Cells have an elongated shape in the direction of their growth, which ensures better compressive strength of foams in this direction [3]. The compressive strength value of the foams containing cork was lower than that of the reference foams in parallel and perpendicular directions. This relationship was correlated with the lower apparent density of the foams modified with the bio-filler. When the apparent density of BPU/CM3 and BPU/CM6 was the same, an increase in compressive strength in the parallel direction was observed. Such an effect can be associated with the lower average cross-sectional area of cells of BPU/CM6 compared to BPU/CM3 (Table 2). There was also a slight effect of a partial replacement of the petrochemical polyol with the bio-polyol on the reduction in the strength of the foam. A similar effect in the case of replacing petrochemical polyol with bio-polyol in the amount of more than 40% by mass was observed in other studies [23]. This is due to the lower hydroxyl number of the bio-polyol in relation to the petrochemical polyol. Given the need to maintain the same value of the isocyanate index, the use of bio-polyol determines a smaller amount of isocyanate added into the PUR composition, which creates rigid segments of the PUR chains responsible for mechanical strength.

The thermal stability of PUR foams is related to both the physical as well chemical structure of the polymer. Increased temperature causes the decomposition of the weakest bonds in the structure of foams, which results in the degradation of PUR materials. A thermogravimetric analysis was performed to study the influence of the cork content on the thermal stability of the porous PUR bio-composites. Table 4 shows the selected results of the thermogravimetric analysis of the foams. Based on the TG curves (Figure 4), the temperatures at which 5%, 20%, and 50% weight loss occurred were determined and marked as T_5%_, T_20%_, and T_50%_, respectively. The amount of solid residue at 900 °C and the maximum temperatures of the individual stages of degradation of the foams (T_1_, T_2_, T_3_) were also determined.

Foams based on the petrochemical polyol were characterized by higher temperatures (~200 °C) at which there was a 5% weight loss compared to the foams based partially on the bio-polyol (~150 °C). With an increase in the filler content, the temperatures at which 5%, 20%, and 50% weight loss occurred also increased for both the PU and BPU foams. An increase in the cork content also increased the amount of solid residue at 900 °C. As shown in other works [36,37], introducing a cork as a filler into the polymeric structure often significantly increased the TGA residue of composite. This effect is related to the cork’s substantial content of lignin and sclereids, characterized by high residual mass fractions at high temperatures [38].

Figure 4 shows the TG and DTG curves of the PUR and composite foams. The PUR foams exhibited a three-stage degradation. The first stage was related to the breakdown of the urethane bonds that formed the rigid segments of the PUR chain. This phenomenon overlaps with the evaporation of water residue from natural filler and the beginning of lignin decomposition. In the second and third stages, the flexible segments degraded. However, in the case of the BPU foams, an additional stage could be distinguished in the temperature of 250 °C, which is associated with the degradation of triethanolamine present in the bio-polyol molecule [39]. The main components of the filler degraded over a wide temperature range. Lignin degraded slowly in the entire measurement temperature range, leaving a high residue content. Hemicellulose and cellulose usually decompose in the temperature range of 220–315 °C and 315–400 °C, respectively [40,41]. The reduced intensity of the peaks on the DTG curves for composites containing increased cork content resulted from the reduced amount of polymer in the matrix and the high temperature of complete suberin decomposition in the filler, reaching 490 °C.

Due to its specific chemical composition, cork is characterized by a high flame resistance resulting from high lignin and suberin contents (waxy substance composed of polyaliphatic and polyaromatic domains interlinked by glycerol) [42]. It creates a specific char when exposed to a flame [41,42]. Thanks to these cork properties, the materials obtained in our study were characterized by a large amount of solid residue after combustion. As reported, the characteristic thermal degradation and response to a flame make cork a filler interacting synergistically with intumescent flame retardants introduced into polymers [43]. PUR foams are flammable materials; therefore, to reduce their flammability, flame retardants and often also fillers are added during synthesis [3]. In our experiment, oxygen index measurements were carried out in order to check the influence of the cork content on the flammability of the foams. The LOI values inform at what oxygen concentration in the oxygen-nitrogen mixture the combustion process of the material is maintained. A test was carried out for the reference foams and the foams containing modified cork in the amount of 6, 9, and 12% by weight (Table 3). The LOI of the materials used in construction should not be lower than 21%, i.e., the approximate oxygen content in the air. All the foams tested in our research were characterized by a higher value of LOI in the range of 21.6–22.4%. For this reason, the foams can be considered materials with reduced flammability. In the case of the foams based on the petrochemical polyol, an increase in the cork content caused an improvement in the composites’ LOI. However, in the case of the foams based partly on the bio-polyol, the addition of cork unexpectedly increased the flammability of the modified foams. Nevertheless, the foams based partly on the bio-polyol were characterized by a higher or similar value of LOI to that of the PU foams.

## 4. Conclusions

Polyurethane materials obtained using renewable raw materials are currently very attractive in various industries, including construction. They fit into the ongoing trend of producing environmentally friendly materials that can increase the energy efficiency of buildings. The research described here demonstrated that it is possible to replace some petrochemical raw materials with raw materials of renewable origin. This was achieved by replacing one of the petrochemical components used for the synthesis of the polyurethane matrix with a waste vegetable oil component. In the case of the PUR systems not modified with bio-polyol, the addition of a cork filler caused a slight decrease in the reactivity of the systems. Such an effect was not observed for systems modified with bio-polyol based on used cooking oil. Additionally, after a successful introduction of a bio-filler, it was found that the thermal insulation properties of the modified biomaterials were comparable to those of the reference material. All the biomaterials obtained in our experiments had good thermal stability at elevated temperatures, which makes them applicable as thermal insulation.

## Figures and Tables

**Figure 1 materials-16-03032-f001:**
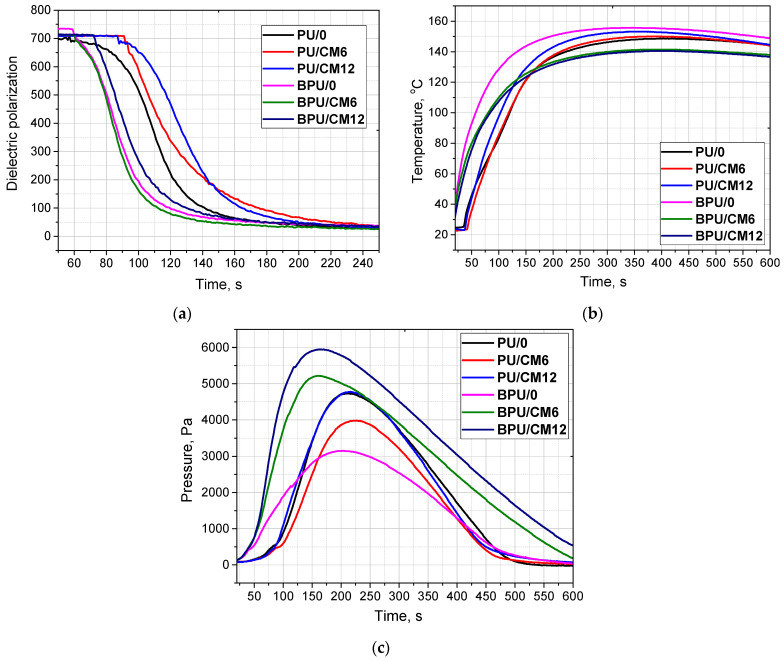
Dielectric polarization (**a**), temperature (**b**), and pressure (**c**) of PUR foams modified with cork vs. the reaction time.

**Figure 2 materials-16-03032-f002:**
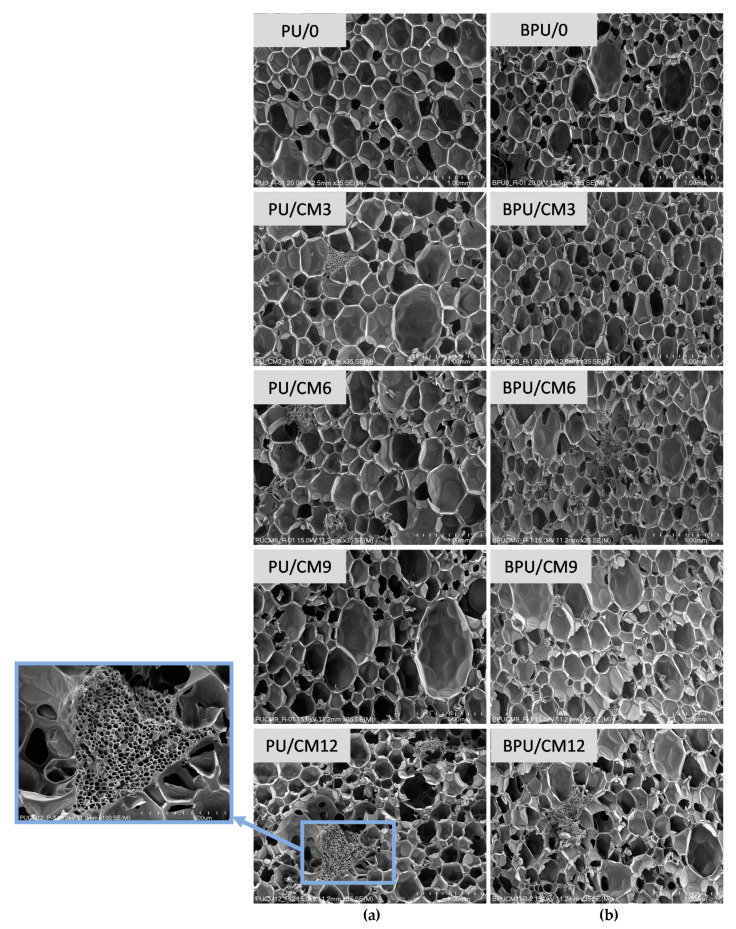
SEM images of cellular structure of foams without (**a**) and with (**b**) the bio-polyol. An exemplary cork particle embedded in the PUR matrix was marked on the SEM photomicrograph.

**Figure 3 materials-16-03032-f003:**
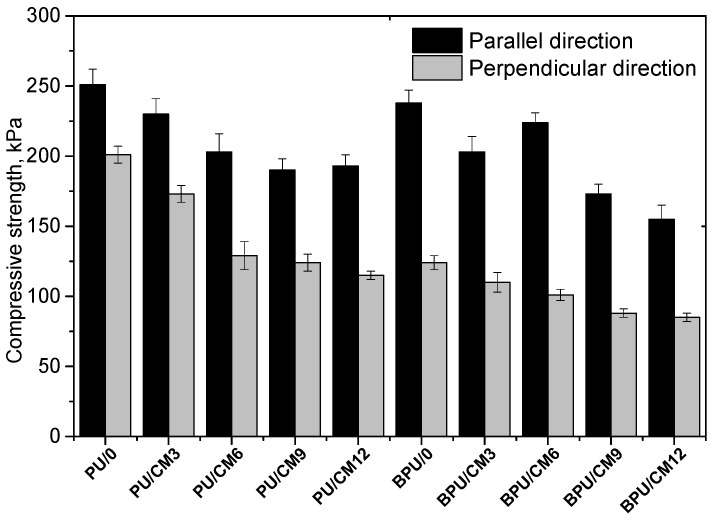
Compressive strength of foams.

**Figure 4 materials-16-03032-f004:**
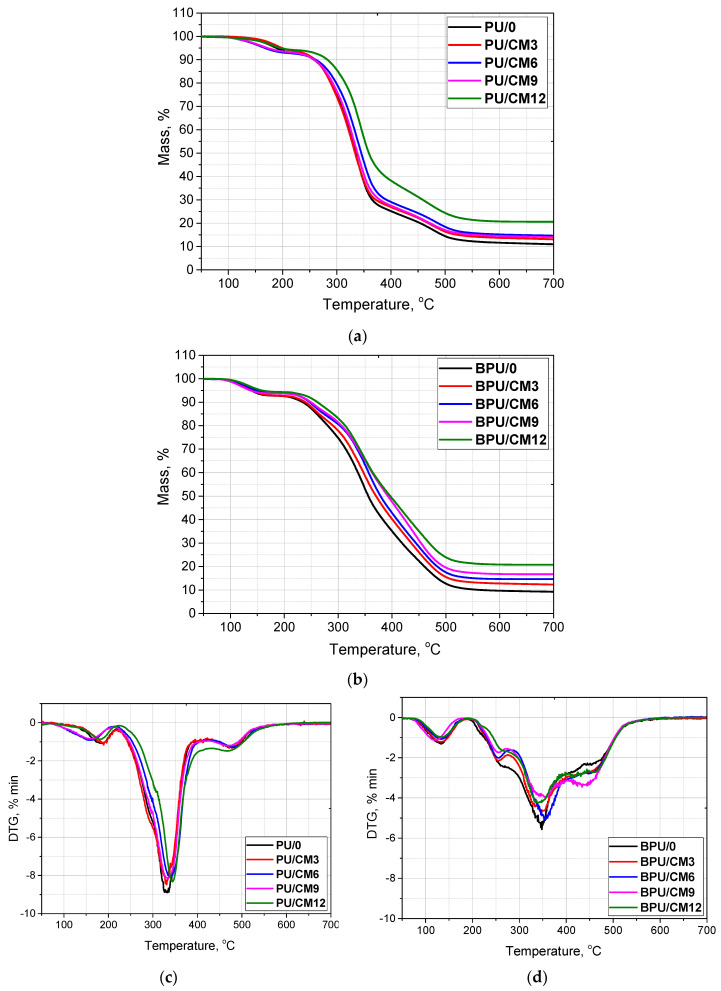
Tg (**a**,**b**) and DTG (**c**,**d**) curves of foams.

**Table 1 materials-16-03032-t001:** Formulations of PUR foams.

Component, g	PU	BPU
Petrochemical polyol	100	50
Bio-polyol	0	50
Catalyst	1.5	1.2
Surfactant	1.5	1.5
Flame retardant	20	20
Water	3	2.8
Cork	0–12	0–12
Isocyanate	162.9	150.9
Content of renewable raw materials in foam, %	0–4	18–21

**Table 2 materials-16-03032-t002:** Characteristic cellular structure parameters of reference and modified foams.

Foam Symbol	The Average Cross-Sectional Area of Cells, mm^2^	Anisotropy Index, -	Content of Closed Cells, %	Apparent Density, kg/m^3^	Coefficient of Thermal Conductivity, mW/m·K
PU/0	0.36	1.33	95.2 ± 1.7	46.7 ± 1.29	25.1 ± 0.01
PU/CM3	0.38	1.30	90.8 ± 2.5	45.4 ± 0.46	26.0 ± 0.14
PU/CM6	0.39	1.35	90.2 ± 1.0	44.4 ± 0.03	26.3 ± 0.13
PU/CM9	0.45	1.39	88.4 ± 1.0	43.9 ± 0.37	26.2 ± 0.08
PU/CM12	0.35	1.35	90.5 ± 2.7	45.2 ± 0.27	25.5 ± 0.35
BPU/0	0.25	1.41	93.6 ± 2.5	46.4 ± 0.22	25.3 ± 0.30
BPU/CM3	0.29	1.41	90.6 ± 2.3	45.5 ± 0.22	25.8 ± 0.13
BPU/CM6	0.27	1.55	89.3 ± 2.8	45.5 ± 0.73	26.4 ± 0.17
BPU/CM9	0.29	1.42	92.0 ± 2.6	44.6 ± 0.41	25.5 ± 0.27
BPU/CM12	0.29	1.41	91.1 ± 1.8	43.6 ± 0.68	26.3 ± 0.32

**Table 3 materials-16-03032-t003:** The selected properties of foams.

Foam Symbol	Water Absorption, %	Water Vapor Diffusion Resistance Factor, -	Brittleness, %	Dimensional Stability, %		
				a	b	c
PU/0	0.61 ± 0.02	15.2 ± 0.73	3.1 ± 0.04	−0.19	−0.24	−0.34
PU/CM3	0.82 ± 0.03	14.6 ± 0.37	2.4 ± 0.05	−0.28	−0.20	0.45
PU/CM6	0.95 ± 0.04	13.7 ± 0.18	1.9 ± 0.48	−0.26	0.01	1.03
PU/CM9	1.15 ± 0.03	13.3 ± 0.46	2.0 ± 0.20	−0.48	−0.58	1.57
PU/CM12	1.37 ± 0.00	13.2 ± 0.51	2.7 ± 0.20	1.19	1.15	0.03
BPU/0	0.76 ± 0.01	20.9 ± 0.92	2.7 ± 0.39	1.21	1.25	−0.15
BPU/CM3	0.78 ± 0.08	18.4 ± 0.41	2.6 ± 0.65	0.20	0.25	0.39
BPU/CM6	0.96 ± 0.00	16.5 ± 1.13	2.2 ± 0.32	0.09	0.08	0.42
BPU/CM9	0.88 ± 0.03	19.1 ± 0.40	2.5 ± 0.32	−0.09	0.14	0.71
BPU/CM12	1.03 ± 0.03	20.3 ± 0.80	2.7 ± 0.29	0.12	−0.01	0.85

**Table 4 materials-16-03032-t004:** Thermal parameters of PU and PU-CM composites obtained from TGA.

Foam Symbol	T_5%_, °C	T_20%_, °C	T_50%_, °C	Residue at 900 °C, %	T_1,_ °C	T_2_, °C	T_3_, °C	LOI, %
PU/0	188.6	290.6	334.2	9.86	186.2	335.6	472.7	21.6
PU/CM3	199.1	288.6	333.6	12.17	188.6	330.3	477.0	21.7
PU/CM6	167.2	298.7	344.9	13.99	157.2	339.7	481.0	21.8
PU/CM9	172.0	292.4	338.0	13.19	159.0	332.4	464.5	21.9
PU/CM12	195.5	315.4	359.0	20.61	180.2	344.0	466.3	22.0
BPU/0	138.1	280.4	355.9	8.68	132.9	347.9	-	22.4
BPU/CM3	141.7	290.1	370.2	11.68	133.3	255.1	347.4	22.2
BPU/CM6	147.4	302.8	378.6	14.63	133.0	256.2	355.3	22.0
BPU/CM9	136.9	306.0	392.4	16.71	119.5	255.0	357.1	21.9
BPU/CM12	158.0	312.7	396.7	20.76	134.9	345.3	456.5	21.0

## Data Availability

Not applicable.

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
