# Peer review of "Cork Porous Biocomposites with Polyurethane Matrix Modified with Polyol Based on Used Cooking Oil"

_materials, 2023, doi:10.3390/ma16083032_

Round 1

Reviewer 1 Report

Kurańska et al. presented a method to prepare renewable porous composite from polyurethane matrix and reused cooking oil as polyol. The experiments were well-designed and the manuscript was prepared in good quality. Before this manuscript is ready for publication. The authors are encouraged to calculate the bio-carbon content of the composite by deriving the molar ratio of bio carbon with respect to total carbon content in your system. For more details of the calculation please refer to ASTM D6866-22 standard.

Author Response

Dear Reviewer,

Thank you very much for the submitted review and comments on the article. We responded to all comments in the answer below. The changes were introduced in the manuscript and marked in color.

KuraÅ„ska et al. presented a method to prepare renewable porous composite from polyurethane matrix and reused cooking oil as polyol. The experiments were well-designed and the manuscript was prepared in good quality. Before this manuscript is ready for publication. The authors are encouraged to calculate the bio-carbon content of the composite by deriving the molar ratio of bio carbon with respect to total carbon content in your system. For more details of the calculation please refer to ASTM D6866-22 standard.

  • Due to the short time to respond to the review, it was not possible to obtain the mentioned standard and calculate the amount of biocarbon. However, we instead calculated the amount of renewable raw materials in the materials obtained.

Reviewer 2 Report

Review of the

Manuscript ID: materials-2271606

Title: Cork porous biocomposites with polyurethane matrix modified with polyol based on used cooking oil

Authors: Maria KuraÅ„ska, Mariusz Ptak, Elżbieta Malewska *, Aleksander Prociak, Mateusz Barczewski, Mateusz Dymek, Fábio A. O. Fernandes, Ricardo
De Sousa, Krzysztof Polaczek, Karolina Studniarz, Katarzyna Uram

The research described in this manuscript proposes to demonstrate that it is possible to replace some petrochemical raw materials with raw materials of renewable origin in various industries.

The manuscript is very interesting, being an important issue in the field. The experimental procedures were well designed.

Paper could be accepted for publication after minor revision, taking into account the following suggestions:

1.      In the Introduction section, the novelty of these experiments should be emphasized

2.      The authors specified :

“ The composites were prepared with 3, 6, 9 and 12 % of cork in relation to the polyol mass of the PU system.”

„The influence of different contents of cork on the changes in dielectric polarization (a), temperature (b) and pressure (c) is shown in Figure 1.”

„Figure 1 Dielectric polarisation (a), temperature (b) and pressure (c) of PUR foams modified with cork vs. the reaction time”

One cannot observe the influence of the cork concentration from these figures. Maybe CM represents cork material?????, and 0, 6, 12 –its concentration?  How the representations change when CM is 3 or 9%?

Please improve the figures to be understandable.

3.      In “Figure 3 Compressive strength of foams.” :

The BPU/CM6 compressive strength value is higher than that of BPU/CM3.

Please explain.

Author Response

Dear Reviewer,

Thank you very much for the submitted review and comments on the article. We responded to all comments in the answer below. The changes were introduced in the manuscript and marked in color.

Paper could be accepted for publication after minor revision, taking into account the following suggestions:

  1. In the Introduction section, the novelty of these experiments should be emphasized
  • At the end of the introduction, information about the innovation of the research subject has been added.

  1. The authors specified :

“ The composites were prepared with 3, 6, 9 and 12 % of cork in relation to the polyol mass of the PU system.”

„The influence of different contents of cork on the changes in dielectric polarization (a), temperature (b) and pressure (c) is shown in Figure 1.”

„Figure 1 Dielectric polarization (a), temperature (b) and pressure (c) of PUR foams modified with cork vs. the reaction time”

One cannot observe the influence of the cork concentration from these figures. Maybe CM represents cork material?????, and 0, 6, 12 –its concentration?  How the representations change when CM is 3 or 9%?

Please improve the figures to be understandable.

  • Analysis on the foamat device was only carried out for selected amounts of cork, as no significant effect of cork addition on temperature, dielectric polarisation and pressure was observed.
  • The drawing has been corrected to make it more clear.

.3.      In “Figure 3 Compressive strength of foams.” :

The BPU/CM6 compressive strength value is higher than that of BPU/CM3. Please explain.

  • When the apparent density of BPU/CM3 and BPU/CM6 was the same, an increase in compressive strength in parallel direction was observed. Such effect can be associated with lower the average cross-sectional area of cells of BPU/CM6 compared to BPU/CM3 (Table 2).

Reviewer 3 Report

Reviewer Comments

Abstract

1.       The keywords material should be added.

2.       The keyword composites should be in the abstract. The biocomposites are suitable that relate to your title.

Introduction

1.       The literature about the utilization of cooking oils is less. Author, please add the information and cite the latest reference. The 2nd paragraph could be replaced with the cooking oil information.

2.       Good introduction about cork material. The author could add information about the availability/abundance of cork. Maybe put some figures and references to support the utilization of cork material.

Experimental

1.       The author should state how much RPM the stirrer uses to prepare the foams.

2.       The paragraph in the 2.2 subtitles has a problem with the reference (Error), please check.

3.       In table 1, there were different values in certain components to produce the foam (Catalyst, water, isocyanate). The author should explain why certain component is different, maybe due to a certain factor/problem to prepare the foam.

Results and Discussion

1.       Figure 1 should be arranged properly.

2.       Figure 2, SEM image should be enlarged. The arrangement could be arranged side by side and could see the difference. The author should mark the SEM image of the different cellular structures of foam biocomposite. The presence of filler also should be marked in the SEM images.

3.       The overall figure needs to be arranged properly. The graph format should be uniform. The author needs to improve this part.

4.       Overall content and findings from this research are good.

Author Response

Dear Reviewer,

Thank you very much for the submitted review and comments on the article. We responded to all comments in the answer below. The changes were introduced in the manuscript and marked in color.

Abstract

  1. The keywords material should be added.
  2. The keyword composites should be in the abstract. The biocomposites are suitable that relate to your title.

  • Keyword “material” was added. The keyword “composite” has been changed to “biocomposites”.

Introduction

  1. The literature about the utilization of cooking oils is less. Author, please add the information and cite the latest reference. The 2ndparagraph could be replaced with the cooking oil information.
  • Literature concerning the use of cooking oil has been added in the second paragraph.
  1. Good introduction about cork material. The author could add information about the availability/abundance of cork. Maybe put some figures and references to support the utilization of cork material.
  • Cork availability information has been added.

Experimental

  1. The author should state how much RPM the stirrer uses to prepare the foams.
  • RPM has been added to the experimental part.
  1. The paragraph in the 2.2 subtitles has a problem with the reference (Error), please check.
  • Instead of the literature reference here, there should be “Table 1”. It was probably removed during editing.

  1. In table 1, there were different values in certain components to produce the foam (Catalyst, water, isocyanate). The author should explain why certain component is different, maybe due to a certain factor/problem to prepare the foam.
  • In the case of the use of a biopolyol, the formulation was modified compared to the formulation containing only the petrochemical polyol. The reduction in the amount of catalyst is due to the greater reactivity of the biopolyol obtained with triethanolamine. The change in the amount of water in the formulation is due to the fact that the biopolyol has a different water content, and the intention was to keep the total amount of water in the system constant. Similarly, the amount of isocyanate was chosen so that the isocyanate index was 1.1 after taking into account the different hydroxyl numbers.
  • An explanation has been added to the text.

Results and Discussion

  1. Figure 1 should be arranged properly.
  • The drawing has been corrected to make it more clear.

  1. Figure 2, SEM image should be enlarged. The arrangement could be arranged side by side and could see the difference. The author should mark the SEM image of the different cellular structures of foam biocomposite. The presence of filler also should be marked in the SEM images.
  • SEM images have been enlarged. The presence of the filler is exemplified in one image.

  1. The overall figure needs to be arranged properly. The graph format should be uniform. The author needs to improve this part.
  • The figures have been corrected.